# House Dust Mite Exposure through Human Milk and Dust: What Matters for Child Allergy Risk?

**DOI:** 10.3390/nu14102095

**Published:** 2022-05-17

**Authors:** Patricia Macchiaverni, Ulrike Gehring, Akila Rekima, Alet H. Wijga, Valerie Verhasselt

**Affiliations:** 1Centre of Research for Immunology and Breastfeeding (CIBF), Medical School and School of Biomedical Science, University of Western Australia, Perth, WA 6009, Australia; akila.rekima@uwa.edu.au (A.R.); valerie.verhasselt@uwa.edu.au (V.V.); 2Immunology and Breastfeeding Group, Neonatal and Life Course Health Program, Telethon Kids Institute, Perth, WA 6009, Australia; 3Institute for Risk Assessment Sciences, Utrecht University, 3508 TC Utrecht, The Netherlands; U.Gehring@uu.nl; 4Centre for Prevention and Health Services Research, National Institute for Public Health and the Environment, 3720 BA Bilthoven, The Netherlands; alet.wijga@rivm.nl

**Keywords:** breastmilk, house dust mite, *Der p* 1, Asthma, total IgE

## Abstract

Allergies are major noncommunicable diseases associated with significant morbidity, reduced quality of life, and high healthcare costs. Despite decades of research, it is still unknown if early-life exposure to indoor allergens plays a role in the development of IgE-mediated allergy and asthma. The objective of this study is to contribute to the identification of early-life risk factors for developing allergy. We addressed whether two different sources of house dust mite *Der p* 1 allergen exposure during early life, i.e., human milk and dust, have different relationships with IgE levels and asthma outcomes in children. We performed longitudinal analyses in 249 mother–child pairs using data from the PIAMA birth cohort. Asthma symptoms and serum total and specific IgE levels in children were available for the first 16 years of life. *Der p* 1 levels were measured in human milk and dust samples from infant mattresses. We observed that infant exposure to *Der p* 1 through human milk was associated with an increased risk of having high levels of serum IgE (top tertile > 150 kU/mL) in childhood as compared to infants exposed to human milk with undetectable *Der p* 1 [adjusted OR (95% CI) 1.83 (1.05–3.20) *p* = 0.0294]. The *Der p* 1 content in infant mattress dust was not associated with increased IgE levels in childhood. The risk of asthma and *Der p* 1 sensitization was neither associated with *Der p* 1 in human milk nor with *Der p* 1 in dust. In conclusion, high levels of IgE in childhood were associated with *Der p* 1 exposure through human milk but not exposure from mattress dust. This observation suggests that human milk is a source of *Der p* 1 exposure that is relevant to allergy development and fosters the need for research on the determinants of *Der p* 1 levels in human milk.

## 1. Introduction

Allergy is an abnormal immune response directed against harmless molecules called allergens. In allergic disorders, such as allergic asthma, the immune response is associated with B cells differentiation that undergo immunoglobulin class-switch recombination to produce IgE antibodies. IgE antibodies are the hallmark of allergic responses and are responsible for acute allergic symptoms upon their cross-linking on mast cells by allergens [1]. Increased levels of IgE can also drive amplification mechanisms in allergic disorders even in the absence of ongoing exposure to specific antigens [1].

Many factors affect the induction of IgE antibodies, including the host genotype, nature of allergen, allergen concentration in the environment and the route of exposure [1]. Recent studies demonstrated that food allergens induce allergic sensitization when in contact with an impaired skin barrier [2], whereas the oral route is thought to favour oral tolerance and allergy protection [3,4,5]. On the other hand, skin [6], respiratory [7] and oral exposure to house dust mite (HDM) allergen, were associated with increased risks of allergic sensitisation, respiratory problems [8] and food allergies in infants [9].

*Dermatophagoides pteronyssinus* is the most widely distributed HDM, and the role of *Der p* 1 allergen in allergic sensitization and disease is largely recognized [7]. Allergic inflammation induced by *Der p* 1 also stimulates allergic responses to bystander allergens [10], thereby contributing to the progression of the atopic march. Although it is usually found in high concentrations in indoor environments [7], the dust is not the only matrix where HDM allergens are found. *Der p* 1 was detected in up to 78% of human milk samples from Australia, France, Brazil and the Netherlands [8,11,12,13]. This observation reinforces that exposure to *Der p* 1 occurs in the first months of life by both respiratory and oral routes.

The design of successful strategies for allergy prevention requires the identification of risk factors in early life; a period that is considered to be a window of susceptibility for the long-term risk of disease [14]. The present work used data from the Prevention and Incidence of Asthma and Mite Allergy (PIAMA) birth cohort to investigate whether the source of *Der p* 1 exposure in early childhood, i.e., human milk versus exposure from mattress dust, influences IgE levels and asthma prevalence throughout the first 16 years of life.

Here, we demonstrated that *Der p*1 in human milk, but not dust, was associated with an increased risk of having high levels of IgE in childhood. This highlights the need to conduct research on *Der p*1 exposure through milk to investigate strategies for allergy prevention.

## 2. Materials and Methods

### 2.1. Study Design and Population

The study population consisted of mother–child pairs from the Prevention and Incidence of Asthma and Mite Allergy (PIAMA) birth cohort, detailed in [15]. In brief, all participants had been followed up since birth by repeated questionnaires at ages 3 months, 1 year, and then annually until age 8, and at ages 11, 14 and 17 years. Medical examinations, including the collection of blood samples for measurements of allergen-specific and total IgE were performed in (different) sub-populations at ages 1, 4, 8, 12 and 16 years. Human milk and dust samples were collected in a sub-group of the population around the infant’s age of 3 months. A total of 249 mother–child pairs with measurements of *Der p* 1 in human milk samples and information on at least one infant health outcome (asthma and/or total IgE) for at least one time point between 4 and 16 years were included. Details on this cohort study were published elsewhere [15].

### 2.2. Samples Collection and Der p 1 Quantification in Human Milk and Mattress Dust

Human milk samples were collected as previously described [16]. *Der p* 1 levels were determined in the aqueous fraction of human milk using an adapted protocol for a high specificity and sensitivity detection, as previously described [8,11,17]. The lower limit of detection (LOD) was 60 pg/mL for twice-diluted human milk samples. Dust samples were collected from infant mattresses according to a standardized protocol, as previously described [18]. Dust extracts were analysed for *Der p* 1 using sandwich enzyme immunoassay (Indoor Biotechnologies, Cardiff, UK). Levels of *Der p* 1 were expressed per gram of dust for samples with detectable amounts of dust (≥11 mg dust). The LOD was 8 ng/mL for 5-fold diluted dust samples. Samples with non-detectable amounts of allergen were assigned a value of two-thirds the detection limit. *Der p* 1 levels in infant mattress dust were available for 198 of the 249 participants.

### 2.3. Health Outcomes: IgE Levels and Asthma

Total and specific IgE levels for *Dermatophagoides pteronyssinus* (*Der p*) were measured by a radioallergosorbent-test-like method (Sanquin Laboratories. Amsterdam, The Netherlands) at ages of 4, 8, 12, and 16 years in children of both allergic and non-allergic mothers [19]. Allergic sensitization was defined as *Der p*-specific IgE ≥ 0.35 kU/L. Parents completed questionnaires for asthma annually until the age of 8 years and then at ages 11, 14, and 16 years. Asthma was defined from age 3 onwards as at least two positive answers to the following three questions: (1) Has a doctor ever diagnosed asthma in your child? (2) Has your child had wheezing or whistling in the chest in the last 12 months? (3) Has your child been prescribed asthma medication during the last 12 months. This definition was developed by a panel of experts within the MeDALL consortium [20].

### 2.4. Statistical Analysis

For all analyses, *Der p* 1 levels in human milk and dust were dichotomized into two categorical variables using the LOD as cut-off values. Total IgE levels were dichotomized into two categorical variables using the top tertile level (>150 kU/L) as cut-off value for “high IgE”. Longitudinal associations of human milk *Der p* 1 levels with high IgE, allergic sensitization, and asthma were assessed by generalized estimation equations with a logit link and compound symmetry within a participant correlation structure. Analyses were performed with and without adjustment for the same covariates as in earlier analyses within the same cohort [19] (child’s age at the time of outcome assessment, maternal asthma and maternal allergy to house dust mites, sex, and pets in the child’s home during the first year of life). Associations of *Der p* 1 levels with allergic outcomes are presented as odd ratios with 95% confidence intervals (CI). Statistical significance was defined by a two-sided a-level of 5%. Calculations were performed using the Statistical Analysis System (SAS 9.4, Cary, NC, USA).

## 3. Results

### 3.1. Characteristics of the Study Population

Characteristics of the study population, such as season of birth, gestational age and duration of breastfeeding are summarized in Table 1. By design, households of mothers with asthma or allergies were over-represented in the study participants when compared with the full PIAMA cohort [15].

### 3.2. Presence of Der p 1 Allergen in Human Milk and Infant Mattress Dust

*Der p* 1 was detected in more than one-third of human milk samples, and the median concentration among these samples was 174 pg/mL (range ≤ LOD-1238 pg/mL, Table 2). We found *Der p* 1 in 41% of dust samples from the infants’ mattresses, and the median concentration among these samples was 1165 ng/g of dust (range ≤ LOD-20,502 ng/g of dust, Table 2).

### 3.3. Associations of Der p 1 Allergen in Human Milk with Child IgE Levels and Asthma

We first analysed levels of total IgE in children breastfed by mothers with detectable versus non-detectable *Der p* 1 in breastmilk. The estimating equation (GEE) analysis of IgE levels at all ages demonstrated that there was a difference of 90.8 kU/mL in IgE levels when comparing children breastfed by mothers with detectable and non-detectable *Der p* 1 in human milk at all ages [95% CI (−6.6; 188.2)], *p* = 0.06. The odds ratio of having high levels of IgE (>150 kU/L) was higher in children exposed to *Der p* 1 in human milk (Figure 1 and Appendix A). Age-specific associations of high IgE levels in serum with the presence of *Der p* 1 in human milk were statistically significant at 12 and 16 years of age (Figure 1 and Appendix A). Overall and age-specific associations persisted after adjustment for confounders (Appendix A). We further assessed whether the prevalence of *Der p* sensitization (Der p spec IgE ≥ 0.35 kU/L) at 4–16 years of age was associated with the presence of *Der p* 1 in human milk. No significant difference was observed when comparing children breastfed by mothers with detectable versus non-detectable *Der p* 1 in human milk (Appendix A). Finally, we analysed whether the presence of *Der p* 1 in human milk was associated with asthma symptoms in the first 16 years of life. A total of 243 children with 1875 observations were included in models with confounder adjustments. The prevalence (Figure 2A) and odds ratios of asthma symptoms tended to increase in children breastfed by mothers with *Der p* 1 in human milk as compared to unexposed children, but associations were not statistically significant [OR (95% CI) 1.38 (0.63–3.04) and 1.47 (0.66–3.27)] for crude and adjusted associations, respectively. The number of asthma cases was small to assess age-specific associations.

### 3.4. Associations of Der p 1 Allergen in Mattress Dust with Child IgE Levels and Asthma

We analysed levels of total IgE in children with detectable versus non-detectable *Der p* 1 in mattress dust collected during the same period of time as breastmilk. When comparing both groups, the mean difference (95% CI) in total IgE levels was −33.7 kU/mL (−146.9; 79.5), *p* = 0.55. The odd ratios of having high levels of IgE (>150 kU/L) were similar in children from both groups (Figure 1 and Appendix A). We then analysed whether the presence of *Der p* 1 in mattress dust was associated with asthma symptoms from 3 to 17 years of life. Although the prevalence (Figure 2B) and odds ratios for asthma symptoms tended to be lower in children with detectable *Der p* 1 in mattress dust compared to those with non-detectable *Der p* 1 in mattress dust [OR (95% CI) 0.79 (0.32–1.95) and 0.63 (0.26–1.52)] for crude and adjusted associations, respectively, no significant association was found. The number of asthma cases was small when assessing age-specific associations.

## 4. Discussion

The lack of knowledge about the sources of allergen exposure in early life hampers the design of successful strategies for the primary prevention of allergies. Here, we confirm previous findings showing that infant mattress and human milk are sources of HDM allergen (*Der p* 1) exposure in early in life [8,9,11,12,13,22]. Importantly, our data suggest an increased risk of having high levels of IgE and a trend of increased asthma risk in children breastfed by mothers with detectable *Der p* 1 in human milk, while such an association was not found for *Der p* 1 in infant mattresses. This observation is consistent with previous findings from the French EDEN birth cohort, where a significantly increased risk of allergic sensitization and respiratory allergies was found in children exposed to *Der p* 1 through human milk [8] but was not related to the proxy of infant exposure to *Der p* 1. The lack of association between *Der p* 1 in dust and infant asthma outcomes is also consistent with previous observations within a larger group of PIAMA participants [19] and other cohorts [23]. This provides evidence that *Der p* 1 exposure though human milk may represent an independent factor that influences infant IgE levels and may set infants on a trajectory towards allergy-prone immunity. The causal relation between *Der p* 1 in breastmilk and increased susceptibility to both respiratory and food allergies was previously confirmed in a mouse pre-clinical study [8,11]. By virtue of their proteolytic activity and capacity to activate pattern recognition receptors, *Der p* 1 allergens induced T helper 2 (Th2) immune responses and triggered allergies to both Der p and bystander allergens. [24]. This may explain why we mainly found an increase in total IgE but not *Der p*-specific IgE.

Interestingly, the odds ratio for having high levels of serum IgE in relation to *Der p* 1 levels in human milk increased with age. Previous studies showed that breastfeeding may convey some protection from asthma during the first years of life, while the benefit would disappear later on [25]. Sears et al. investigated the long-term outcomes of asthma and atopy in breastfed and non-breastfed infants and demonstrated that breastfeeding increased the risk of allergic sensitization to common respiratory allergens from 13 to 21 years and doubled the risk of asthma from 9 years until adulthood [26]. Another study that investigated the long-term effects of exclusive breastfeeding on allergic outcomes in a cohort of 8583 children showed that breastfeeding was protective against asthma at 7 years of age, but this risk was reversed by the age of 14 years, becoming a consistent risk factor until 44 years in infants from atopic mothers [27]. The mechanisms underlying such a long-term association of breastfeeding with developing allergies are currently unknown. We can speculate that *Der p* 1 in human milk may set infants on a trajectory where subsequent gene–environment interactions will lead to the development of allergies.

Even though the total level of IgE was shown to be an important predictor of asthma in population-based studies [28,29] a limitation of our study is the relatively small sample size and consequently the small number of children with asthma outcomes. Moreover, because of the small sample size, we could account for a limited set of potential confounders only (the same as in previous analyses for consistency) [19]. We therefore cannot rule out residual confounding. This prevents us from drawing a firm conclusion for the relevant sources of *Der p* 1 exposure regarding asthma risk, and this stresses the need for larger prospective studies to replicate our findings.

In conclusion, *Der p* 1 exposure through human milk was associated with high levels of IgE in children, while exposure through mattress dust was not. This observation may explain the current lack of consensus on the role of early-life mite allergen exposure in infant allergic symptoms and markers [23]. Since earlier findings suggest that *Der p* 1 content in the indoor environment is not related to the presence of *Der p* 1 in human milk [22], human milk should be included as an independent source of early-life *Der p* 1 exposure in research investigating strategies for allergy prevention.

While there is no doubt that breastfeeding is the most powerful way to prevent infectious diseases and reduce child mortality [30], its effects on allergy prevention are controversial [30,31,32]. By identifying which factors in breastmilk are protective for allergy and learning how to modulate them, we will make breastmilk more able to prevent allergic diseases.

## Figures and Tables

**Figure 1 nutrients-14-02095-f001:**
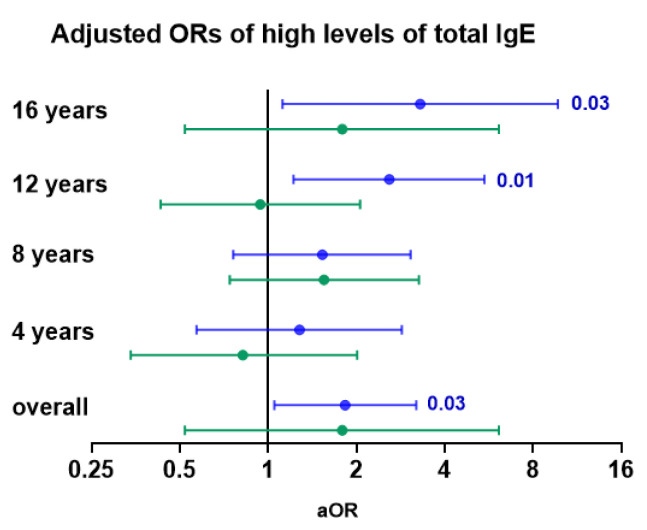
Adjusted associations of high IgE levels in children with *Der p* 1 levels in human milk and infant’s mattress dust. Adjusted odds ratios (aORs) for high levels of IgE (top tertile, > 150 kU/L) at 4–16 years of age in children exposed to detectable versus non-detectable *Der p* 1 in human milk (blue circles) and in infant’s mattress dust (green circles). ORs were adjusted for child’s age at the time of reporting asthma and maternal asthma, as well as for maternal allergy to house dust mites, sex, and pets in the child’s home during the first year of life.

**Figure 2 nutrients-14-02095-f002:**
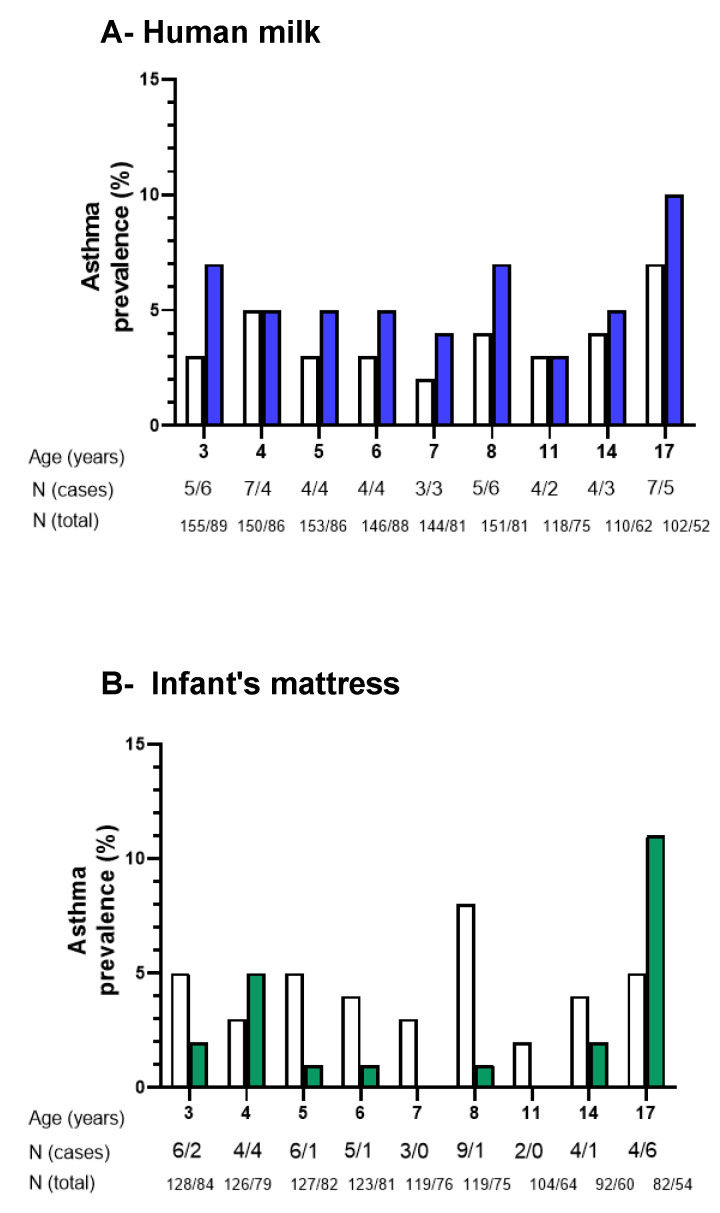
Prevalence of asthma according to the presence of *Der p* 1 in human milk and infant’s mattress. Prevalence of asthma in the past 12 months in 3–17-year-old children exposed to non-detectable (empty columns) and detectable (filled column) *Der p* 1 in human milk (**A**) and mattress dust (**B**). N represents the number of children with asthma symptom (cases) and the total population (total). Fisher’s exact test was used to compare the prevalence of asthma between groups.

**Table 1 nutrients-14-02095-t001:** Characteristics of the study population.

	Study Participants
Infant male sex	126/249 (51%)
Maternal asthma, n/N (%)	40/249 (16%)
Maternal allergy to house dust (mites), n/N (%)	95/244 (39%)
Season of birth, n/N (%)	
Winter	85/249 (34%)
Spring	80/249 (32%)
Summer	49/249 (20%)
Autumn	35/249 (14%)
Maternal age at birth, mean (SD)	31.1 (3.8), n = 244
Gestational age at birth, mean (SD)	40.1 (1.4), n = 249
Infant birth weight (grams), mean (SD)	3532 (492), n = 249
Caesarian section, n/N (%)	20/248 (8%)
Infant age at human milk collection (days), mean (SD)	108 (29), n = 245
Duration of breastfeeding (weeks), mean (SD)	30 (13), n = 249
Pets in the child’s home during 1st year, n/N (%)	104/248 (42%)

Maternal history of allergies and/or asthma was defined from a validated self-reported questionnaire [15,21].

**Table 2 nutrients-14-02095-t002:** Distribution of *Der p* 1 levels in human milk and dust samples collected from child’s mattress.

*Der p* 1 Detection	Human Milk (pg/mL)	Dust from Infant’s Mattress (ng/g)
Number of samples	218	198
Samples with detected *Der p* 1 (%)	79 (36)	81 (41)
Median * (concentration range)	174 [<LOD-1238]	1165 [<LOD-20,502]

* For ≥ LOD samples. Detection limit 60 pg/mL and 8 ng/ mL for human milk and dust, respectively.

## Data Availability

Not applicable.

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
