# Peer review of "House Dust Mite Exposure through Human Milk and Dust: What Matters for Child Allergy Risk?"

_nutrients, 2022, doi:10.3390/nu14102095_

Round 1
Reviewer 1 Report
It is interesting study in which the authors assessed association between exposure to dust mite through from milk and dust and allergy risk. This is perspective study which was helpful to understand long impact of the earlier exposure to mite of human on later health. However, due to smaller sample size, especially for each age intervals, the conclusion should be cautious. There are some technical issues which should be addressed further if a revision is required.
- It is unclear on follow-up, especially for sample size variation during follow-up. From supplementary table 1, it seems that just half of participants were followed. Authors should specify this issue in methods.
- For figure 1, why did authors select those age groups to present these association?
- Analysis in figure 2 should be cautious because of smaller sample size for each age group.
- A key point in this analysis is confounder adjustment. It is suggested that authors should present consideration on this adjustment in the methods, and also present limitation on this issue in the part of discussion.
Author Response
REVIEWER 1
It is interesting study in which the authors assessed association between exposure to dust mite through from milk and dust and allergy risk. This is perspective study which was helpful to understand long impact of the earlier exposure to mite of human on later health. However, due to smaller sample size, especially for each age intervals, the conclusion should be cautious. There are some technical issues which should be addressed further if a revision is required.
We thank the reviewer for the carefully analysis of our manuscript and constructive suggestions.
- It is unclear on follow-up, especially for sample size variation during follow-up. From supplementary table 1, it seems that just half of participants were followed. Authors should specify this issue in methods.
We thank the reviewer for pointing this out. The differences in samples size result from the fact that all participants have been followed up since birth by repeated questionnaires, but that medical examinations including collection of blood samples for measurements of IgE have been performed in a subsets only. We now clarify this as follows (lines 72-76 of the marked up version of the revised manuscript):
“In brief, all participants have been followed up since birth by repeated questionnaires at ages 3 months, 1 year and then annually until age 8, and at ages 11, 14 and 17 years. Medical examinations including collection of blood samples for measurements of allergen specific and total IgE were performed in (different) sub-populations at ages 1, 4, 8, 12 and 16 years.”
- For figure 1, why did authors select those age groups to present these association?
Age 1 year measurements were excluded as at age 1 blood samples and IgE measurements were available for children of allergic mothers only (by design) while at all other ages, blood samples were available for children of both, allergic and non-allergic mothers. In the study we included all ages at which IgE measurements have been performed in children from both allergic and non-allergic mothers.
We now clarify this in line 97 of the marked version of the revised manuscript as follows: “Total and specific IgE levels for Dermatophagoides pteronyssinus (Der p) were measured by a radioallergosorbent test-like method (Sanquin Laboratories. Amsterdam, The Netherlands) at the child’s ages of 4, 8, 12, and 16 years in children of both, allergic and non-allergic mothers.”
- Analysis in figure 2 should be cautious because of smaller sample size for each age group.
We agree with the reviewer’s comment and therefore clarify in the results section that the number of cases are small (lines 158 and 170 of the marked version of the revised manuscript):
“The number of asthma cases was small to assess age-specific associations”.
We also acknowledge this limitation in the discussion (lines 222-224 of the marked version of the revised manuscript): “Even though total level of IgE was shown to be an important predictor of asthma in population-based studies[28, 29] a limitation of our study is the relatively small sample size and consequently the small number of children with asthma outcomes.”
- A key point in this analysis is confounder adjustment. It is suggested that authors should present consideration on this adjustment in the methods, and also present limitation on this issue in the part of discussion.
We describe the confounder adjustment in the statistical analysis part of the methods section (lines 111-115 of the marked version of the revised manuscript): “Analyses were performed with and without adjustment for the same covariates as in earlier analyses within the same cohort[19] (child’s age at the time of outcome assessment, maternal asthma and maternal allergy to house dust mites, sex, and pets in the child’s home during the first year of life).” It is also indicated in figure 1 and Supplementary table 1 legends.
In addition, we added the following text to the discussion section (lines 224-227 of the marked version of the revised manuscript): “Moreover, because of the small sample size, we could account for a limited set of potential confounders only (the same as in previous analyses for consistency) consistency [Gehring U et al., Allergy. 2012]. We therefore cannot rule out residual confounding”.
Reviewer 2 Report
The authors provide an interesting paper on the effects of two different sources of exposure to house dust mites in early childhood on total IgE levels, sensitization to house dust mites and asthma in longitudinal perspective. Survey was properly designed and manuscript is well written. The strengths of the study is longitudinal observation on the basis on the well-known PIAMA bith cohort. The final results are that high levels of IgE in childhood are associated with Der p 1 exposure through human milk but not exposure from mattress dust and these both exposures are not influencing the risk of sensitization to house dust mites and asthma. The interesting question would be if this elevated level of total IgE in the population fedded with human milk with Der p1 presence is influencing the presence of any clinical symptoms of allergy. In terms of primary prevention the results are interesting, confirms that the way of exposure to allergens matters. But they are also a bit controversial as , as far as I known, we can not control the presence of house dust mites in human milk, so the message about the higher risk for allergies by breastfeeding can be discouraging for mothers.
Anyway these observations are very interesting.
Minor comment: In line 226 the is a lack of citation (ref.)
Author Response
The authors provide an interesting paper on the effects of two different sources of exposure to house dust mites in early childhood on total IgE levels, sensitization to house dust mites and asthma in longitudinal perspective. Survey was properly designed, and manuscript is well written. The strengths of the study is longitudinal observation on the basis on the well-known PIAMA bith cohort. The final results are that high levels of IgE in childhood are associated with Der p 1 exposure through human milk but not exposure from mattress dust and these both exposures are not influencing the risk of sensitization to house dust mites and asthma. The interesting question would be if this elevated level of total IgE in the population fedded with human milk with Der p1 presence is influencing the presence of any clinical symptoms of allergy
We are very pleased to read this paper gathered your interest and we would like to thank the reviewer for his/her positive feedback.
We agree with the reviewer that it would be interesting to analyse if high levels of total IgE in infants is influencing the risk of any clinical symptoms of allergy, but this study was not designed to analyse that question. However, our previous publication in mouse models and birth cohorts studies highlighted a possible role for Der p 1 in breastmilk as a risk factor for food and respiratory allergies (see introduction lines 47-49 and discussion lines 202-204)
In terms of primary prevention, the results are interesting, confirms that the way of exposure to allergen matters. But they are also a bit controversial as, as far as I know, we can not control the presence of house dust mites in human milk, so the message about the higher risk for allergies by breastfeeding can be discouraging for mothers. Anyway, these observations are very interesting.
We agree with the statement regarding a possible higher risk of allergies associated with breastfeeding might discourage mothers from breastfeeding. Given this concern, we revised our conclusion (lines 237-241 of the marked version of the revised manuscript) as follow:
“While there is no doubt that breastfeeding is the most powerful way to prevent infectious diseases and reduce child mortality [31], its effects on allergy prevention are controversial [31-33]. By elucidating which factors in breastmilk are protective or not for allergy and how to modulate them, we will make breastmilk more likely to prevent allergic diseases.”
Minor comment: In line 226 the is a lack of citation (ref.)
We thank the reviewer for pointing this out. The reference has been added.[Macchiaverni, P., et al., House dust mites: Does a clean mattress mean Der p 1-free breastmilk? Pediatr Allergy Immunol, 2020. 31(8): p. 990-993].
Round 2
Reviewer 1 Report
The authors have addressed most of my comments.